# The Soluble Urokinase-Type Plasminogen Activator Receptor as a Biomarker for Survival and Early Treatment Effect in Metastatic Colorectal Cancer

**DOI:** 10.3390/cancers13205100

**Published:** 2021-10-12

**Authors:** Kristian Blomberg, Torben F. Hansen, Claus L. Brasen, Jeppe B. Madsen, Lars H. Jensen, Caroline B. Thomsen

**Affiliations:** 1Department of Health Science, University of Southern Denmark, 5000 Odense, Denmark; krblo15@student.sdu.dk; 2Danish Colorectal Cancer Center South, Department of Oncology, University Hospital of Southern Denmark, 7100 Vejle, Denmark; Torben.Hansen@rsyd.dk (T.F.H.); Lars.Henrik.Jensen@rsyd.dk (L.H.J.); 3Department of Regional Health Research, University of Southern Denmark, 5000 Odense, Denmark; Claus.Lohman.Brasen@rsyd.dk; 4Department of Biochemistry and Immunology, Lillebaelt Hospital, University Hospital of Southern Denmark, 7100 Vejle, Denmark; jeppe.buur.madsen@rsyd.dk; 5Open Patient Data Explorative Network, The Clinical Institute, Odense University Hospital, 5000 Odense, Denmark

**Keywords:** suPAR, metastatic colorectal cancer, monitoring

## Abstract

**Simple Summary:**

Patients with metastatic colorectal cancer are likely to have a poor outcome. Chemotherapy is the primary treatment, but because of toxicity and a lack of benefit for a large group of patients, it is important to monitor the treatment effect. The only tool for evaluating treatment effect currently is imaging. Biomarkers associated with disease progression could therefore be important for treatment monitoring in the future. The biomarker suPAR is prognostic for the overall survival in colorectal cancer. This study aimed to investigate if plasma suPAR levels before and during treatment were prognostic of survival and predictive for treatment effect. Our study confirmed that higher plasma suPAR levels before initiation of palliative chemotherapy were prognostic of shorter survival.

**Abstract:**

The soluble urokinase-type plasminogen activator receptor (suPAR) is prognostic for overall survival (OS) in colorectal cancer (CRC). Our study explored the association between baseline suPAR and OS and progression-free survival (PFS) in metastatic CRC (mCRC). It is also the first study to explore the association between the initial change in suPAR level and OS, PFS and the first CT response evaluation. The study included 132 patients with mCRC treated with chemotherapy (FOLFIRI) with or without an EGFR-inhibitor. Blood samples were drawn before the first treatment cycle and in between the first and second treatment cycle. suPAR levels were determined using an ELISA assay. Using the Kaplan-Meyer method, we demonstrated a significantly shorter OS for patients with suPAR levels above the median (HR = 1.79, 95%CI = 1.10–2.92, *p* = 0.01). We also showed association between plasma suPAR level, gender and performance status (PS). However, we could not show any association with PFS, and analysis on the change in suPAR level provided no significant results. The results showing association between baseline suPAR and OS are in line with previous findings.

## 1. Introduction

In 2020 it was estimated that >1.9 million new cases and 935,000 deaths of colorectal cancer (CRC) occurred worldwide. It is the third most common type of cancer, accounting for 10% of new cases [1]. At the time of diagnosis 15–25% of CRC patients have metastatic disease and another 25% develop it later [2,3]. The prognosis has improved due to better treatment and earlier detection [4,5], however, outcomes are still poor for patients with metastatic disease.

It is essential to give the right treatment, but equally important to discontinue ineffective treatment. This makes monitoring a central issue. Prognostic biomarkers may improve the overall evaluation of the disease and treatment course and therefore hold the potential to improve the quality of life for CRC patients.

An extracellular protease system, the urokinase-type plasminogen activator (uPA) system, is active in cancer invasion and dissemination [6]. The urokinase-type plasminogen activator receptor (uPAR) is a protein attached to the cell membrane by a glycosyl-phosphatidylinositol anchor and consists of three domains [7]. The uPAR have been shown to be upregulated in cancer, both on the invasive front and in the tumor core, on macrophages and on some cancer cells [8,9,10,11]. It has also been shown to be prognostic for overall survival (OS) in CRC and other gastrointestinal cancers [12,13,14,15,16,17,18].

The uPA can cleave uPAR on the cell surface between domain I and II on the cell surface, liberating domain I (uPAR(I)). Both intact and cleaved uPAR (uPAR(I-III) and (II-III), respectively) can be shed from the cell surface through proteolytic activity. Three soluble uPAR (suPAR) forms can therefore be measured in the blood; uPAR(I), suPAR(I-III) and suPAR(II-III) [7]. Studies have shown that an increased concentration of suPAR in plasma may be of diagnostic importance both in patients with symptoms of CRC, and patients with non-specific symptoms and signs of other cancers [19,20].

Previously it has been shown that measuring the plasma and serum levels of suPAR, especially the combined amount of suPAR types, before any treatment is given (surgery and/or chemotherapy) is prognostic for OS in CRC patients [21,22,23,24,25,26]. This association may be due to a casual association between suPAR and the physiology of the cancer. It may, however, also be due to suPAR being increased due to a range of other diseases like cardiac disease [27], kidney disease [28] and infections [29], since suPAR is an unspecific marker of disease presence [30]. In cases where cancer treatment has an effect and association with the change in suPAR can be established, suPAR may be used for monitoring cancer treatment.

To our knowledge, this is the first study to explore the relationship between a change in plasma suPAR levels during chemotherapy and outcomes in CRC patients. It is important to identify whether patients are benefitting from their ongoing regimen.

The aim of this study was to explore if plasma suPAR levels at baseline in patients with mCRC treated with 5-flourouracil, folinic acid and irinotecan (FOLFIRI) is prognostic for OS and progression-free survival (PFS), and if the dynamics of suPAR between baseline and the first treatment cycles holds additional information.

## 2. Materials and Methods

The study was designed as a phase III biomarker study (adapted from Wagner 2012 [31]) to determine how well suPAR predicts clinical outcomes by testing the marker against specimens collected longitudinally from a research cohort. Patients with mCRC treated at the Department of Oncology, Vejle Hospital, Denmark, were offered inclusion in a prospective biomarker study. Inclusion criteria were histopathological verified adenocarcinoma of the colon and rectum, metastatic disease, planned treatment with FOLFIRI, performance status (PS) 0–2, life expectancy ≥3 months, evaluable disease according to the Response Evaluation Criteria in Solid Tumors (RECIST) 1.1 [32], age 18 years or above and written and orally informed consent to participation.

Plasma samples were collected at baseline and between the first and second treatment cycle, with a few exceptions collected between the second and third treatment cycle (*n* = 4). All patients received treatment with FOLFIRI. Additionally, patients with wild-type RAS/RAF received treatment with an EGFR-inhibitor. The treatment effect was evaluated after the fourth cycle of chemotherapy according to RECIST 1.1. A change in plasma suPAR levels was defined as an increase or decrease of 10% or more, A lesser change was classified as stable. The study is presented in accordance with the Reporting recommendations for tumour marker prognostic studies (REMARK) [33].

Peripheral venous blood samples were collected into EDTA containing tubes. Plasma samples were separated by centrifugation at 2000× *g* for 10 min within 2 h following venipuncture and stored until use at minus 80 degrees Celsius. Plasma levels of suPAR(I-III) and suPAR(II-III) were analyzed by the suPARnostic^TM^ ELISA assay (Virogates A/S; Birkerød, Denmark) in a batch to minimize analytical variation and according to the manufacturer’s instructions. This assay does not measure the plasma levels of uPAR(I). In the presentation of our results suPAR(I-III) and suPAR(II-III) will be referred to as suPAR. The results were reported in μg/L and were preferentially included in statistical analyses as a continuous variable. There is no established clinically validated cut-off, but the results were dichotomized by a median spilt to allow simple analysis and illustrations [34].

Descriptive statistics are presented by their median and range of continuous variables, number and percentage of categorical ones.

Survival was calculated from the date of written and oral consent of inclusion in the study to the date of death of any cause, using the Kaplan-Meyer method to estimate survival probabilities and Cox regression analysis to estimate risk ratios (RR). In the same way, PFS was calculated to the date of reported radiologically progressive disease (PD). PFS was censored at the date the patient were allocated to a follow-up program without treatment. OS was censored at the last known date alive with a data cut off in April 2021. The dynamics of suPAR between the two blood samples were tested for significant difference in OS and PFS using the Kaplan-Meyer method for the three groups: stable, increasing and decreasing. To test for significant association in response to treatment between the merged groups, stable/decreasing and increasing, the Fisher exact test was performed. The two categories of response for this were PD/stable disease (SD) and partial response (PR)/complete response (CR). To test for association between the characteristics of the study population and OS, we performed simple and multiple Cox regression analysis. Results from the multiple Cox regression analysis are only shown for parameters that were statistically significant in the simple analysis.

All statistical analyses were performed using NCSS 2011 Statistical Software (NCSS, LLC. Kaysville UT, USA). *p*-values < 0.05 were considered statistically significant and all reported *p*-values were two-sided.

Information on survival, progression and characteristics were collected from the patients’ medical records. All patients were included prospectively, and data were analyzed retrospectively.

## 3. Results

### 3.1. Patient Characteristics

Between November 2015 and February 2021, 153 patients were enrolled in the study. Blood samples were not available for analysis in 20 patients, leaving 133 patients for investigation. One patient was treated again after a break and was enrolled twice. Only the blood samples from the first enrollment were included in the study. One patient withdrew consent to participate in the protocol while still being in active treatment. For this patient, PFS was censored on the day of withdrawal, while OS was followed until the last known date alive in the patients’ medical records. The second blood sample was unavailable for analyses in two cases. Data from these two patients are included in the baseline OS and PFS analyses, but excluded from the dynamics analyses. The same approach was applied to two patients with baseline and follow up blood samples showing plasma suPAR levels above the upper measurement level of ELISA (>13.83 μg/L). For the Fisher exact test there was one additional patient censored as the patient stopped treatment after the second cycle of FOLFIRI with no evaluation CT scan. Table 1 shows the characteristics of the 132 patients included in the study. There were statistically significant association between baseline suPAR level and PS and gender. Female gender and PS 1–2 was associated with higher suPAR levels at baseline. The highest median suPAR level was found in patients with microsatellite instability (MSI) tumors. No correlation was found between the baseline plasma suPAR level and the location of primary tumor, RAS/RAF-status, MSI-status, age, M-stage or the number of prior treatment regiments.

### 3.2. Prognostic and Predictive Value of suPAR

The median OS for the entire population was 17.4 months (95%CI = 15.5–19.7 months). Survival analysis showed that patients with plasma suPAR levels above the median (4.25 μg/L, range: 2.29–>13.83 μg/L) had significantly shorter survival compared to patients with levels below the median (HR = 1.79, 95%CI = 1.10–2.92, *p* = 0.01), as shown in Figure 1a. Median PFS for the entire population was 7.9 months (95%CI = 7.2–7.9). Plasma suPAR levels above the median showed no prognostic significance for PFS when compared to plasma suPAR levels below the median (HR = 1.31, 95%CI = 0.68–2.52, *p* = 0.42), Figure 1b.

The dynamics of suPAR between baseline and the second blood sample showed no statistically significant difference for OS (*p* = 0.45) nor PFS (*p* = 0.69), shown in Figure 2. The Fisher exact test showed no statistically significant association in response to treatment between stable/decreasing or increasing groups (Table 2), with *p* = 0.73.

### 3.3. Cox Regression Analysis

In our simple Cox regression analysis for OS (Table 3), the variables age, gender, M-stage at diagnosis, PS, tumor location, MSI-status, plasma suPAR level, line of treatment and RAS/RAF status were included. Of these, PS (RR = 3.21, 95%CI = 1.94–5.31, *p* ≤ 0.01) and plasma suPAR levels (RR = 1.76 95%CI = 1.06–2.92, *p* = 0.03) were significantly prognostic for OS. In the multiple analyses, PS (RR = 3.05, 95%CI = 1.83–5.07, *p* ≤ 0.01) was still independently significant, while plasma suPAR levels (RR = 1.59, 95%CI = 0.95–2.65, *p* = 0.08) were not.

## 4. Discussion

In this prospective study of mCRC patients, we aimed to explore the association between plasma suPAR levels at baseline and OS and PFS. To our knowledge, this is also the first study to explore the association between the change in plasma suPAR level from baseline after the first cycle of chemotherapy and OS, PFS and the response to treatment. Our study showed a statistically significant association between baseline plasma suPAR levels above the median and shorter OS.

In the present study population, we observed a statistically significant association between median plasma suPAR levels at baseline, gender and PS. Higher median plasma suPAR levels (total suPAR and suPAR(I-III)) in females is something that also has been observed in other studies of CRC patients [24,25,26]. This is also the case for PS, where Tarpgaard et al. [21] observed a significantly higher plasma suPAR level in patients with PS 1–2 when compared with PS 0, which is similar to our findings. The literature is still sparse on the relation of suPAR levels to CRC patient characteristics, but there seems to be a tendency towards those of an older age being weakly, but significantly, related to higher suPAR levels [22,24,25,26]. This was not observed in our study population. Furthermore, contrary to our study, it also seems as if the location of the primary tumor is associated to plasma suPAR level, with patients presenting with tumors in the colon having significantly higher levels than patients with tumors in the rectum [21,25]. We could not find any association between plasma suPAR levels at baseline and PFS. Unlike our study, Tarpgaard et al. [21] showed an association between baseline suPAR levels (suPAR(I-II) + (II-III)) and PFS in their univariate analysis (*p* ≤ 0.01). However, in their multivariate analysis there was no significance for PFS.

In a simple cox regression analysis, our study demonstrated a significant association between plasma suPAR levels at baseline and OS. Levels above the median were associated with reduced OS. This is in agreement with several other studies on CRC patients [16,21,22,23,24,25,26], confirming the prognostic value of baseline suPAR levels. However, unlike other studies, suPAR was not independently significant for OS in the multiple cox regression analysis of our study population adjusting for PS. We also observed that PS at the time of inclusion was significantly prognostic for OS, both in the simple and multiple cox regression analysis.

The results did not demonstrate any significant relationship between the dynamics of suPAR and OS, PFS or response at the first evaluation CT scan. Rolff et al. [23] showed that plasma suPAR (suPAR(I-III) + (II-III) and uPAR(I)) levels 6 months after surgery in CRC patients were significantly associated with survival in their multivariate analysis (*p* ≤ 0.01, respectively). The change in suPAR levels from before surgery was included as a covariable and were independently significant (*p* ≤ 0.01, respectively). This demonstrates that the dynamics of suPAR may hold potential to be of prognostic value. Our study aimed to explore this potential in a population consisting of mCRC patients, and the potential clinical value of evaluating the effect of treatment at an early stage. Currently there is no biomarker available for clinical use to monitor progression during treatment of mCRC patients. This leaves imaging as the only recommended modality of evaluation [35]. CT scanning has its shortcomings, as it is not suitable for small metastases and the need for a minimum of 20% increase in size according to the RECIST 1.1 criteria. Given the serious toxicity of chemotherapy, it should hold high priority to find a biomarker that would make it possible to discontinue ineffective treatment at an earlier stage than at present. In this regard, blood-based biomarkers hold an advantage over markers in tissue, as they can be measured repeatedly with easy access. Different biomarkers have emerged in mCRC patients. Through decades CEA has been discussed with various clinical value. Recently circulating tumor DNA (ctDNA) has been greatly evaluated in both the localized and metastatic settings of these patients. In a clinical setting a “reference change value” (RCV) indicating the minimum significant change between two samples would be of interest if the biomarker is to be used in treatment monitoring. The RCV value takes both the analytical and biological variances of suPAR into account. At the moment, the RCV of suPAR is not known, but could be assumed to be between 20% and 40%. With the criteria of 20% change, our population would have had three patients with increasing plasma suPAR levels, and 27 patients with decreasing levels. This gave a significant result in difference in OS (*p* ≤ 0.01). The value of this, however, is limited with so few patients in the increasing group. The difference with regards to PFS was still insignificant (*p* = 0.13). The RCV may show to decrease over time, as the analytical laboratory capabilities improv, and the biology is better understood to account for parameters of possible preanalytical influence. We therefore decided to use 10% change as our criteria to generate a basis for future studies. The median plasma suPAR level at baseline in our study was higher than what has been observed in other studies that used ELISA methods [22,24]. It has previously been shown that high uPAR levels in the tumor are associated with metastatic disease [18]. It is possible that our population has higher levels of uPAR, given that they have metastatic disease. However, the median plasma suPAR levels in our study are also higher than the median level of patients with metastatic disease in the two cited studies. It is important to add that we have not explored the uPAR levels in our population, and, therefore, it is not certain that the uPAR levels in our population would correspond with previous findings. Another possible explanation is that different methods may influence the results. Studies on the diagnostic use of suPAR levels in kidney disease have suPAR measurements that are difficult to compare across studies. One study comparing two widely used suPAR ELISA kits showed significantly different cut off values for diagnosis [36]. This demonstrates the importance of standardized or comparable methods across studies. At the very least, the importance of being aware of these differences when comparing results. suPAR can be measured through various analytical methods (ELISA, turbidimetric, proteomic approaches and Point-Of-Care-Testing) all of which have advantages and disadvantages in terms of turn-around time, analytical variation and levels of results. We chose the ELISA assay from Virogates A/S as it, to date, is the only ELISA assay which is approved for clinical use, not only for research use. In case suPAR is found useful for mCRC patients, switching to a turbidimetric assay would be beneficial due to the low analytical variance and faster turn-around time [37]. Virogates A/S is also the only available supplier of turbidimetric assays for suPAR. Standardization would be necessary in order to compare studies using difference analytical assays as shown by Hayek et al. [38].

This study has several limitations. Firstly, we used the median plasma suPAR level as the cut off point. This is an arbitrary limit. Ours and other studies have shown that suPAR hold prognostic value. However, a biological cut off point needs to be established for it to be of more practical clinical value. A biological cut off point like this is not yet known and future studies in this area are warranted. Further analysis of ROC-curves (for death after 180 days (Appendix A), 270 days (Appendix A), 365 days (Appendix A) and 530 days (Appendix A)) after the main analysis of this study (see Appendix A) suggest that a potential cut off for future studies could either be 5.0 μg/L, or 3.6 and 6.0 μg/L. The latter would create three groups that respectively could be analyzed though Kaplan-Meier. Another potential limitation of our study is that we chose to use a percentage of change to divide our population into groups of stable, increasing and decreasing plasma suPAR levels. It takes a greater change in level to reach the predefined 10% for the patients with higher baseline levels compared to the ones with lower levels. Perhaps a value of absolute change would have been a more appropriate way to investigate this for future clinical application, however, this is out of scope for this study.

## 5. Conclusions

We observed that baseline plasma suPAR levels were prognostic for OS in a population of CRC patients in accordance with previous studies. We could not show significant association between the dynamics of suPAR after one cycle of FOLFIRI with or without an EGFR-inhibitor and response at the first evaluation CT scan. Further studies on suPAR are justified to investigate the possible value in subgroups, to validate proposed cut off thresholds and to test clinical outcomes in mCRC prospectively.

## Figures and Tables

**Figure 1 cancers-13-05100-f001:**
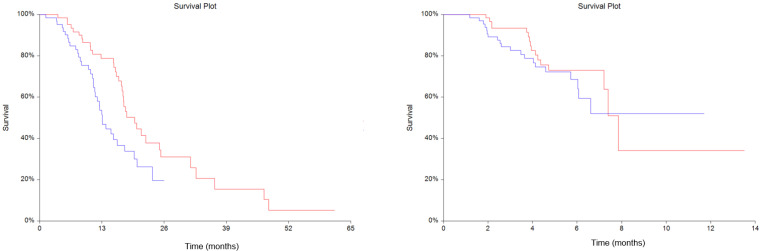
Kaplan-Meyer estimates of survival probabilities in overall survival (OS) (**a**) (Hazard ratio (HR) = 1.79, 95% Confidence interval (CI) = 1.10–2.92, *p* = 0.01) and progression free survival (PFS) (**b**) (HR = 1.31, 95%CI = 0.68–2.52, *p* = 0.42). Red line: plasma soluble urokinase-type Plasminogen Activator Receptor (suPAR) levels below the median. Blue line: plasma suPAR levels above the median.

**Figure 2 cancers-13-05100-f002:**
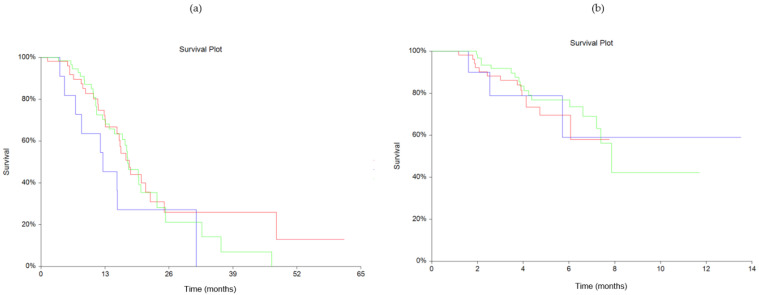
Kaplan-Meyer estimates of survival probabilities relating to changes in plasma soluble urokinase-type Plasminogen Activator Receptor (suPAR) levels for overall survival (OS) (**a**) (*p* = 0.45) and progression free survival (PFS) (**b**) (*p* = 0.69). Red line: stable plasma suPAR levels. Blue line: increasing plasma suPAR levels. Green line: decreasing plasma suPAR levels.

**Table 1 cancers-13-05100-t001:** Demographic characteristics of the 132 patients with baseline suPAR included in the study.

Parameter	*n* (%)	Median Baseline suPAR Level (μg/L)	*p*-Value
Median age, year (range)	65.9(41.3–81.1)	-	-
>65	70(53%)	4.43	0.21
≤65	62(47%)	4.22	-
Gender			
Female	44(33.3%)	4.80	**0.03**
Male	88(66.7%)	4.22	-
Location of primary tumor			
Right/transverse colon	42(31.8%)	4.34	0.78
Left colon	43(32.6%)	4.68	-
Rectum	46(34.8%)	4.23	-
Unknown	1(0.8%)	3.91	-
M-Stage at diagnosis			
1	91(68.9%)	4.32	0.74
0	41(31.1%)	4.35	-
MSI-status			
MSS	125(94.7%)	4.26	0.44
MSI	5(3.8%)	5.91	-
Unknown	2(1.5%)	5.12	-
Prior treatment regiments			
0	77(58.3%)	4.41	0.06
≥1	55(41.7%)	3.91	-
Performance status			
0	93(70.5%)	4.01	**<0.01**
1–2	39(29.5%)	5.50	-
RAS/RAF-status			
Mutated	60(45.5%)	4.31	0.99
Wild Type	72(54.5%)	4.34	-

Statistically significant results are bold.

**Table 2 cancers-13-05100-t002:** Overview of the groups and categories of the 127 patients with two blood samples. The Fisher exact test resulted in *p* = 0.73.

Status	Dynamics		Total
	Increase	Stable/Decrease	
Progressive disease/stable disease	9	82	91
Partial or complete response	2	34	36
Total	11	116	127

**Table 3 cancers-13-05100-t003:** Cox regression simple and multiple analyses, overall survival, *n* = 132.

Parameter	Simple Test		Multiple Test	
	RR (95% CI)	*p*-Value	RR (95% CI)	*p*-Value
Gender				
Female	1.08 (0.66–1.78)	0.76	-	-
Male	1	-	-	-
Age, median 70				
>65	1.08 (0.67–1.76)	0.74	-	-
≤65	1	-	-	-
M-stage at diagnosis				
1	0.81 (0.48–1.37)	0.43	-	-
0	1	-	-	-
Performance status				
1–2	3.21 (1.94–5.31)	**<0.01**	3.05 (1.83–5.07)	**<0.01**
0	1	-	1	-
Tumor location				
Rectum	0.67 (0.37–1.22)	0.19	-	-
Left colon	0.74 (0.41–1.34)	0.32	-	-
Right colon	1	-	-	-
MSI status				
MSS	1.51 (0.37–6.18)	0.57	-	-
MSI	1	-	-	-
suPAR				
≥ median	1.76 (1.06–2.92)	**0.03**	1.59 (0.95–2.65)	0.08
< median	1	-	1	
Line of treatment				
Second or later	1.40 (0.87–2.26)	0.17	-	-
First	1	-	-	-
RAS/RAF status				
Mutated	1.04 (0.62–1.76)	0.87	-	-
Wt	1	-	-	-

Statistically significant results are bold. Variables included are gender, age, M-stage at diagnosis, performance status, tumor location, microsatellite instability (MSI) status (instable (MSI) or stable (MSS)), the plasma level of the soluble urokinase-type Plasminogen activator receptor (suPAR), line of treatment and RAS/RAF status (Kirsten rat sarcoma virus, Neuroblastoma rat sarcoma virus viral oncogene homolog, proto-oncogene B-raf) mutated or Wild type (Wt). Risk Ratio (RR) with confidence intervals (CI) and p-value are shown for both the simple and multiple test.

## Data Availability

The dataset supporting the conclusions of this article is not available due to danish data protection legislation.

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
