# Peer review of "The Soluble Urokinase-Type Plasminogen Activator Receptor as a Biomarker for Survival and Early Treatment Effect in Metastatic Colorectal Cancer"

_cancers, 2021, doi:10.3390/cancers13205100_

Round 1
Reviewer 1 Report
Maybe extend the discussion part, but it is a very interesting study, you should include more patients or do it as a multicenter study.
Author Response
Reviewer 1: "Maybe extend the discussion part, but it is a very interesting study, you should include
more patients or do it as a multicenter study."
• Thank you for the comment. We will consider a prospective multicenter study but as this is a
retrospective study, it is difficult to extend the patient population based on different
methodologies on different centers.
Reviewer 2 Report
The authors address a topic of great interest and the performed experiments are appropriate. The authors demonstrated that in mCRC patients there is an statistically significant association between suPAR high level expression and OS.
In the discussion the authors should write a summary of has been reported in literature on recently emerged biomarkers for mCRC patients.
The authors in the discussion argue about the different methodologies used for the uPAR plasma detection and the threshold used (eg ELISA). Given the interest of this point, the authors should focus on this point and illustrate the different possibilities for quantification with regard to advantages and disadvantages.
Author Response
The authors address a topic of great interest and the performed experiments are appropriate. The
authors demonstrated that in mCRC patients there is a statistically significant association between
suPAR high level expression and OS.
In the discussion the authors should write a summary of has been reported in literature on recently
emerged biomarkers for mCRC patients.
• We thank the reviewer for the comment and have added a short describtion of the recently
emerged biomarkers in mCRC patients (see page 7, line 231-234).
The authors in the discussion argue about the different methodologies used for the uPAR plasma
detection and the threshold used (eg ELISA). Given the interest of this point, the authors should
focus on this point and illustrate the different possibilities for quantification with regard to
advantages and disadvantages."
• Which analytical method is truly important in case of early biomarker research since the
different methods have yet to be standardized and therefore results can vary a lot.
We have added a short description of why we chose ELISA and about potential future use of
a turbidimetric assay (see page 8, line 259-268)
Reviewer 3 Report
Blomberg et al. investigated the prognostic association between plasma suPAR levels and OS & PFS in
mCRC patients who had chemotherapy treatment. They also measured changes in suPAR levels during
the course of chemo-treatment. The study collected blood samples before and during the treatments
from 132 late staged mCRC patients, and the plasma suPAR levels (ELISA against suPAR domains I-III
and II-III) were measured with a single commercial ELISA kit. The study observed a subgroup of
patients who had higher plasma suPAR levels (above the median) had significantly shorter OS
compared to a subgroup of patients with lower suPAR levels (below the median) - no prognostic
significance was observed for PFS. Additionally, no significant differences were observed for neither
OS nor PFS when associated with the level of uPAR changes during the treatment.
There are several limitations in this study (e.g., arbitrary suPAR cut off level), however, the authors
are aware of the limitations and stated in the manuscript.
Although some of the authors’ findings were contrary to other previous studies (e.g., the prognostic
association between suPAR and PFS), it is important to report conflicting results to the field which will
improve the understanding of the prognostic values of usPAR.
Overall, the study was appropriately designed and relevant statistical methods were used. Some of
the authors’ findings are already observed from previous studies, however, the focus of the current
study has clinical significance as there is no biomarker available to monitor progression during the
treatment of mCRC patients.
Additional points to be discussed:
Prognostic efficacy of plasma/serum suPAR (as well as uPAR in tumours, PMID: 25692297) in CRC has
been demonstrated by a large number of studies for more than 20 years. However, the suPAR/uPAR
has not been translated into the clinic yet. Some researchers carefully assert that the chance of
translational outcomes of uPAR is uncertain. It is an important clinical question in the
prognostic/diagnostic biomarker discovery fields as the question is not only for uPAR. As the authors
suggested “Further studies on suPAR and the potential prognostic and predictive value in mCRC are
warranted”, the manuscript should have an additional discussion on why suPAR/uPAR is yet to be
translated into the clinic.
It is interesting to observe that the plasma suPAR levels were associated with shorter OS before
initiation of palliative chemotherapy, but not during the course of treatment. It would be very
interesting to know how overall plasma proteome changes during treatment. Mass Spectrometry
(MS)-based proteome would be a good approach to understand the overall plasma proteome changed
(or alterations in other specific protein(s) as a biomarker discovery). The manuscript would be
improved if the authors discuss more about what will be further studies to improve the understanding
of the proteome alterations in mCRC during chemotherapy treatment(s).
Since CRC is a more common disease for older ages, associating the OS may provide false-positive
results. If available, it would be better to use disease-free survival (DFS) data (suggestion only).
As stated by authors (in lanes 250-251), “different methods may influence the results”, different uPAR
ELISA kits may provide different results due to antibody variations. It would be good to validate the
results with orthogonal methods such as MS-based targeted approach proteomics (suggestion only –
not requesting to perform additional experiments).
Minor points:
Lane 132: remove duplicated words “a break”
Table 1: add a unit for Median Baseline suPAR level
Author Response
Blomberg et al. investigated the prognostic association between plasma suPAR levels
and OS & PFS in mCRC patients who had chemotherapy treatment. They also
measured changes in suPAR levels during the course of chemo-treatment. The study collected blood
samples before and during the treatments from 132 late staged mCRC patients, and the plasma
suPAR levels (ELISA against suPAR domains I-III and II-III) were measured with a single
commercial ELISA kit. The study observed a subgroup of patients who had higher plasma suPAR
levels (above the median) had significantly shorter OS compared to a subgroup of patients
with lower suPAR levels (below the median) - no prognostic significance was observed for PFS.
Additionally, no significant differences were observed for neither OS nor PFS when associated with the level of uPAR changes during the treatment.
There are several limitations in this study (e.g., arbitrary suPAR cut off level), however, the authors
are aware of the limitations and stated in the manuscript.
Although some of the authors’ findings were contrary to other previous studies (e.g., the prognostic
association between suPAR and PFS), it is important to report conflicting results to the field which
will improve the understanding of the prognostic values of usPAR.
Overall, the study was appropriately designed and relevant statistical methods were used. Some of
the authors’ findings are already observed from previous studies, however, the focus of the current
study has clinical significance as there is no biomarker available to monitor progression during the
treatment of mCRC patients.
Thank you for the valuable comments.
Additional points to be discussed:
Prognostic efficacy of plasma/serum suPAR (as well as uPAR in tumours, PMID: 25692297) in
CRC has been demonstrated by a large number of studies for more than 20 years. However, the
suPAR/uPAR has not been translated into the clinic yet. Some researchers carefully assert that the
chance of translational outcomes of uPAR is uncertain. It is an important clinical question in the
prognostic/diagnostic biomarker discovery fields as the question is not only for uPAR.
As the authors suggested “Further studies on suPAR and the potential prognostic and predictive
value in mCRC are warranted”, the manuscript should have an additional discussion on
why suPAR/uPAR is yet to be translated into the clinic.
This is an important yet difficult question to answer. Most studies are retrospective and
prospective studies are missing in general. Prospective testing and validation are central
elements in order to elucidate the clinical utility. Standadized testing and cut-offs are
equally important and represent the primary obstacles for now. Since many factors can
influence on suPAR level, e.g. operations, infections, serious/acute illness and smoking, there may be subgroups of mCRC patiens in which suPAR has good predictive value and
other subgroups with poor predictive value.
It is interesting to observe that the plasma suPAR levels were associated with shorter OS before
initiation of palliative chemotherapy, but not during the course of treatment. It would be very
interesting to know how overall plasma proteome changes during treatment. Mass Spectrometry
(MS)-based proteome would be a good approach to understand the overall plasma proteome
changed (or alterations in other specific protein(s) as a biomarker discovery). The manuscript would
be improved if the authors discuss more about what will be further studies to improve the
understanding of the proteome alterations in mCRC during chemotherapy treatment(s).
Thank you for the comment. Proteome alterations in mCRC during chemotherapy treatment
is truly an interesting subject which has great promise to help understand the biology of
resistance to chemotherapy. High-resolution mass-spectrometry and the complexity hereof
is unfortunately out of scope for this manuscript.
Since CRC is a more common disease for older ages, associating the OS may provide false-positive
results. If available, it would be better to use disease-free survival (DFS) data (suggestion only).
Thank you for the suggestion. In metastatic disease, as this study investigate, it does not
seem reasonable to define disease-free survival, as none of the patients are disease-free.
As stated by authors (in lanes 250-251), “different methods may influence the results”, different
uPAR ELISA kits may provide different results due to antibody variations. It would be good to
validate the results with orthogonal methods such as MS-based targeted approach proteomics
(suggestion only – not requesting to perform additional experiments).
Again, thank you for the comment and suggestion. Unfortunately, it is not possible for us to
validate the results with MS-based targeted approach proteomics. Certainly, the different
analytical methods should be standardized before entering into clinical use. However, only
the method we used, is currently marketed for clinical use (for emergency medicine).
Minor points:
Lane 132: remove duplicated words “a break”
Table 1: add a unit for Median Baseline suPAR level"
• These have been corrected.